

# The importance of model horizontal resolution for improved estimation of Snow Water Equivalent in a mountainous region of Western Canada

Samaneh Sabetghadam[1,2], Christopher G. Fletcher[2] and Andre Erler[3]

[1] Institute of Geophysics, University of Tehran, Tehran, Iran
[2] Department of Geography & Environmental Management, University of Waterloo, Ontario, Canada
[3] Aquanty Inc., Waterloo, Ontario, Canada

*Correspondence to*: Samaneh Sabetghadam (ssabet@ut.ac.ir)





**Abstract.** Accurate estimation of snow water equivalent (SWE) over high mountainous regions is essential to support water resource management. Due to the sparse distribution of in situ observations, models have been used to estimate SWE. However, the influence of horizontal resolution on the accuracy of simulations remains poorly understood. This study evaluates the potential of the Weather Research and Forecasting (WRF) model at horizontal resolutions of 9, 3 and 1 km to estimate the daily values of SWE over the mountainous South Saskatchewan River Basin (SSRB) in Western Canada, for a representative water year, 2017-18. Results show an accumulation period from October 2017 to the annual peak in April 2018, followed by a melting period to the end of water year. All WRF simulations tend to underestimate annual SWE, with largest biases (up to 58 kg/m$^2$, i.e. relatively 24%) at higher elevations and coarser horizontal resolution. The two higher-resolution simulations capture the magnitude (and timing) of peak SWE very accurately, with only a 3 to 6% low bias for 1 km and 3 km simulations, respectively. This demonstrates that a 3 km resolution may be appropriate for estimating SWE accumulation across the region. A relationship is identified between model elevation bias and SWE biases, suggesting that the smoothing of topographic features at lower horizontal resolution leads to lower grid cell elevations, warmer temperatures, and lower SWE. Overall, high resolution WRF simulations can provide reliable SWE values as an accurate input for hydrologic modeling over a sparsely monitored mountainous catchment.

## 1 Introduction

On average, almost 65% of Canada's landmass is covered by annual snow cover for more than six months of the year (ECCC, 2022). Melting snow in spring is a critical component of the water cycle to determine water supplies and flood risk; however, estimating the effect of snowmelt on flooding depends on a reliable estimate of snow water equivalent (SWE) (Dozier et al. 2016; Wrzesien et al. 2017; Vionnet et al., 2020). SWE is defined as the product of snowpack depth and bulk density and is a key environmental variable for understanding climate (Brown et al., 2019). It represents the vertical depth of water that would be obtained if all the snow cover melted completely (WMO, 2018). The value of SWE shows the amount of liquid water, which is produced from a melting snowpack.

Reliable and accurate estimation of SWE has been significantly required to improve management and analyses of water resources. It is also essential for other applications, including global snow hydrology, global change analysis, and risk assessment (Taheri and Mahmoudian, 2022). Spatiotemporal distribution of SWE, particularly within northern latitudes and higher elevations, shows the extent of spring and summer runoff (Barnet et al., 2005; King et al., 2020). Due to the sparse distribution of in situ observations globally, regional weather forecast models are recently used to estimate the amount of SWE (Klehmet et al., 2013; Wrzesien et al. 2017; Raparelli et al., 2021). Although there have been many studies that evaluate temperature and precipitation simulations over North America (Diaconescu et al., 2016; Xu et al., 2019; Holtzman et al., 2020), few studies have been performed regionally to validate the model estimation of the spatial and temporal patterns in SWE using ground-based observation (e.g., Alonso-González et al. 2018; Mortezapour et al., 2020, Yang et al. 2023).

There is ample evidence from previous studies that model horizontal resolution is one of the key factors that should be improved to increase the accuracy of a simulated snowpack (Leung et al. 2003). Regional climate simulations using a coarse horizontal grid spacing typically underestimate the snowfall compared to the observations. One specific example showed that reducing MM5 model (fifth-generation Penn State–NCAR Mesoscale Model) grid spacing to 13 km led to



an improved estimation of the snowpack for the western United States (Leung and Qian, 2003). Garvert et al. (2007) found that a horizontal resolution of less than 4 km in a high-resolution mesoscale model is required to appropriately simulate the snowfall over a complex terrain and to produce updraft and downdrafts that had a significant impact on the snowfall patterns. In another study, WRF model simulations at 2-km grid spacing for the Colorado Rocky Mountains are analyzed by Rasmussen et al. (2011). The estimations are verified using Snowpack Telemetry (SNOTEL) data. Their results show that the model successfully simulated spatial and temporal patterns of SWE over the region.

The Rocky Mountains in the USA and Canada stretch from the northernmost part of western Canada, to the northern New Mexico in the southwestern United States. The eastern slopes of the Canadian Rocky Mountains, is a complex region and several factors such as season, vegetation, and topography, control the discharge of headwater streams from high elevation catchments to valley bottoms (Hauer et al., 1997). Our study region comprises the eastern foothills region of the Rocky Mountains and the mountain headwaters region of the South Saskatchewan River Basin (SSRB) (see Fig. 1) and it has been more focused on the western SSRB region, which includes mountainous areas of the SSRB.

The SSRB in Western Canada is a major agricultural basin of Canada with a semi-arid climate and highly dependent on surface water (Martz et al. 2007) which mainly comes from the spring snowmelt (Tanzeeba and Gan, 2012). The SSRB is a major sub-basin of the Nelson River Basin of Canada, rising from the Rocky Mountains in the west and extending eastward through southern Alberta (Tanzeeba and Gan, 2012). The watershed has a sub humid to semiarid continental climate. Temperatures can reach 40°C during the summer and −40°C during winter (Martz et al. 2007). During the wintertime, precipitation is principally in the form of snow. Most of the annual runoff (around 70%) of the rivers in this region is supplied from the Rocky Mountains and the foothills (Ashmore and Church 2001). Annually SSRB accounts for nearly 57% of the total water allocated in Alberta. The importance of snow in this region, where supply appears insufficient for projected needs, is elaborated in the literature (Pentney and Ohrn, 2008: Hopkinson et al., 2012). The surface water supply in SSRB region mainly comes from the spring snowmelt (Tanzeeba and Gan, 2012), which makes it highly suitable to study the variability of SWE and its potential hydrologic impact. The objective of the current study is to evaluate the potential of the Weather Research and Forecasting (WRF) model run at fine horizontal resolution to simulate the daily values of snow water equivalent (SWE) over the SSRB. This has direct relevance to hydrological applications.

In-situ observation of snow using the Canadian historical Snow Water Equivalent (CanSWE) dataset is used to evaluate the potential of WRF to detect the monthly variability in SWE. Particular attention is paid to investigate the applicability of WRF model on the accurate estimation of peak SWE and monthly SWE changes. The SWE values also have been compared to the ERA5 and ERA5-Land ERA5 and ERA5-Land reanalysis data to emphasize on the role of resolution. Finally, the impact of elevation has been examined by evaluating several diagnostics. This study can provide information related to the regional water management and hazard prevention.

This paper is organized as follows. Section 2 includes the details about the study region as well as introduction on WRF model, ERA5, ERA5-Land and CanSWE dataset. Section 3.1 analyze the area-averaged temporal evaluation of WRF SWE and quantifies bias and errors between WRF, ERA5, ERA5-Land and CanSWE dataset throughout the study period. Section 3.2 present WRF SWE spatial evaluations for individual stations using statistical metrics, so can provide insights into probable impact of elevation on biases, which is studied in section 3.3. A summary and conclusions are provided in Sect. 4.



## 2 Data and Methodology

### 2.1 CanSWE dataset

In-situ observation of SWE has been widely used in many applications including water and flood forecasting, climate studies, and evaluation of numerical weather prediction models. SWE can be measured manually or automatically as the mathematical product of snow depth and density. The methods that widely are used to measure SWE include, snow cores, snow pits and snow pillows (Elder et al., 1998; Andreadis and Lettenmaier, 2006; Dixon and Boon, 2012). Snow pits and snow courses are manual methods and rely on interpolation to characterize snow depth. This may lead to some errors if snow depth is variable (Lopez-Moreno et al., 2011). However, Snow pillows, measuring SWE by weighing the mass of a snow column, are the most common automatic method for continuous monitoring of SWE at a fixed location. It provides valuable time series of snow, despite they are spatially sparse and expensive to install and maintain (Johnson and Marks, 2004).

The Canadian historical Snow Water Equivalent dataset (CanSWE) combines manual (snow surveys) and automated (includes snow pillows and passive gamma sensors) pan-Canadian SWE observations (Vionnet et al., 2021). This new dataset replaces the Canadian Historical Snow Survey (CHSSD) dataset (Brown et al., 2019) by correcting the metadata, removing duplicate observations and controlling the quality of the records. In Canada, the majority of in situ SWE measurements are collected by provincial or territorial governments and hydropower companies and their partners. CanSWE dataset was compiled from 15 different sources and includes SWE information for all provinces and territories that measure SWE from 2607 locations across Canada over the period from 1928 to 2020. More details on this dataset are provided by Vionnet et al. (2021).

Table 1 shows the location of stations selected to evaluate WRF model performance. All stations were located over the area represented by the innermost WRF model domain. These stations have been chosen based on the availability of daily data during the study period. Daily SWE values are obtained from automated snow pillow stations to better evaluate the results. The evaluation was conducted from 1st October 2017 to 1st October 2018, as the 2018 water year. Our preliminary investigation shows that the 2018 water year had approximately average SWE values during 1984 to 2021 according to the CanSWE stations. Therefore, 2018 can be a representative of the region's climate over the past 38 years. Statistical metrics were considered to evaluate the model simulations against CanSWE data: Pearson correlation coefficient (R), mean bias (MB) and root-mean squared error (RMSE). The evaluation has been done for each station as well as the aggregate of the stations by examining SWE timeseries and its annual and spatial distribution.

**Table 1.  Location of the stations from CanSWE in British Columbia and Alberta.**

| Station name | Station Number | Lattitude | Longtitude | Elevation (m) | Province |
|---|---|---|---|---|---|
| Wild Cat Creek | 1 | 51.70 | -116.63 | 2122 | British Columbia |
| Skoki Lodge | 2 | 51.54 | -116.06 | 2120 | Alberta |
| Floe Lake | 3 | 51.05 | -116.13 | 2090 | British Columbia |
| Sunshine Village | 4 | 51.08 | -115.78 | 2230 | Alberta |
| Three Isle Lake | 5 | 50.63 | -115.28 | 2160 | Alberta |
| Little Elbow Summit | 6 | 50.71 | -114.99 | 2120 | Alberta |
| Mount Oldum | 7 | 50.49 | -114.91 | 2060 | Alberta |
| Lost Creek South | 8 | 50.17 | -114.71 | 2130 | Alberta |
| Soth Racehorse Creek | 9 | 49.78 | -114.60 | 1920 | Alberta |





## 2.2 WRF model configuration

The Weather Research and Forecasting (WRF) model was developed by the National Center for Atmospheric Research (NCAR) to support both operational weather forecasting and atmospheric research. Detailed documentation of WRF model can be found in Skamarock et al. (2008). In this study, the Advanced Research WRF (ARW) version 4.3.2 is used with three one-way nested domains, each with progressively finer horizontal resolution. The outer domain has a resolution of 9 km and covers most of western Canada (Fig.1). The middle domain, with a resolution of 3 km, extends over British Columbia and parts of Alberta. The innermost domain has the highest horizontal resolution of 1 km and

covers the western part of the Southern Saskatchewan River Basin (Fig.1). This version of WRF runs with 38 vertical levels between the Earth's surface and a model top at 50 hPa, which is the same for all domains. The initial and lateral boundary conditions are derived from the 3-hourly and $0.25^{o}$ resolution ERA5 reanalysis (Hersbach et al., 2020) from the European Centre for Medium-Range Weather Forecasts (ECMWF). Simulation results are output on a 6-hour time step, which is aggregated to daily frequency for direct comparison with observations.

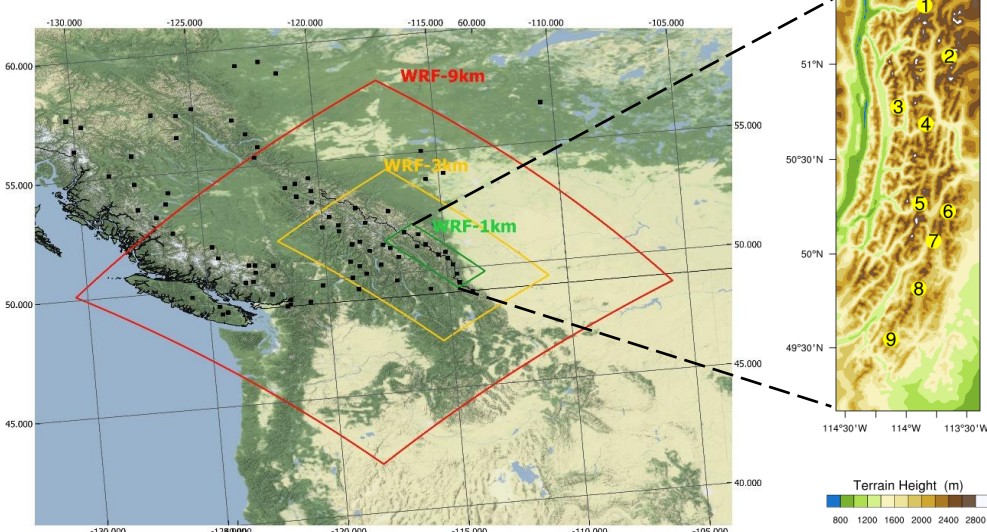

**Figure 1. WRF model domains over Western Canada and the terrain height for the inner domain. The outer boundary of the 9km (red), 3km (yellow) and 1km (green) domains are indicated by the rectangles. Black squares indicate the location of CanSWE automated stations in British Columbia and Alberta. The topography of the 1km domain is shown magnified on the right with the CanSWE stations from Table 1 indicated in the yellow circles.**

The physical parameterization schemes are selected based on previous studies that employ the WRF model to evaluate the simulation of terrestrial snow accumulation over the northern hemisphere (e.g., Niu et al., 2011; Wrzesien et al. 2015; Liu et al., 2017; Brandth et al., 2020; Li and Li, 2021). In particular, the Thompson et al. (2008) cloud microphysics scheme, the rapid radiative transfer model longwave scheme (Mlawer et al., 1997), the Dudhia shortwave scheme (Dudhia, 1989), the Yonsei University planetary boundary layer scheme (Hong et al., 2006), the modified

Kain–Fritsch convective parameterization for the outer domain (Kain and Fritsch, 1990, 1993; Kain, 2004), and the Noah LSM with multi-parameterization (Noah-MP) option (Niu et al., 2011) are used here. Previous studies show that Noah-MP simulates snow more accurately at finer resolution than previous versions of the Noah land surface model (e.g., Wrzesien et al. 2015). Simulated values were extracted at the nearest grid-cell corresponding to the location of



each station, assuming that the in-situ observation is representative of a model gridded area. It is acknowledged that

such point comparisons of SWE are inherently challenging due to the heterogeneity in elevation, aspect and land cover (Cui et al., 2023). However, given the reasonably representative station density within the innermost WRF domain (Fig.1), we attempt to mitigate these issues by also comparing simulated and observed spatial mean SWE using a spatial mean taken over all stations.

The evaluation of WRF results in the current study has been focused on the discussion on innermost domain that

includes the eastern foothills region of the Rocky Mountains and the mountain headwaters region of the SSRB (Fig. 1). We emphasize that this innermost domain is simulated at all three resolutions; in other words, at each resolution the model produces output over its entire domain, not just the outer part.

**2.3 ERA5 and ERA5-L**

The datasets used in this study also included ERA5 and ERA5-Land (hereafter, ERA-L), to explore the consistency of

the ERA5 and ERA5-L reanalysis datasets in the SWE estimation. As mentioned in section 2-2, the 0.25o resolution ERA5 reanalysis has been also used as the initial and lateral boundary conditions for WRF run. ERA5 is the fifth generation ECMWF atmospheric reanalysis (Hersbach et al., 2020) and has a grid resolution of 31 km. This is higher resolution than in the older ERA-Interim of 80 km. ERA5 is based on advanced modeling and data assimilation systems, i.e. the Integrated Forecasting System (IFS) Cycle 41r2, and combines large amounts of historical observations

into global estimates. It provides hourly fields for all variables. ERA5 assimilates snow properties from several SYNOP stations, and from year 2004 onwards, it also uses IMS data over NH (Hersbach et al., 2020). On the other hand, ERA5-L is the land component from ERA5 with a finer spatial resolution of 9 km. It is produced with land model H_TESSEL and without coupling the atmospheric module without data assimilation (Muñoz-Sabater et al., 2021). These reanalysis data are used to evaluate their SWE values and to understand the role of resolution in SWE estimation over the region.

**3. Results**

**3.1 Evaluation of the spatial mean SWE**

The time series of daily SWE values from CanSWE, ERA5, ERA5-L and WRF simulations averaged over the inner domain of the SSRB are presented in Figure 2. The seasonal cycle of SWE in observations shows a clear accumulation period from Oct 1 to peak SWE (648 mm) in late April, and a melting period from late April until late June. In general,

this seasonal evolution of the snowpack is well represented by the WRF simulations at all three resolutions as well as the reanalysis ERA5 and ERA5-L. However, in agreement with previous studies (e.g., Wrzesien et al. 2018), our results confirm that the reanalysis products significantly underestimate mountain SWE. The peak SWE occurs on the same day, 22nd April, for all WRF resolutions and the CanSWE observations, indicating that the WRF model is conserving the primary details of the meteorological lateral boundary forcing required for snow accumulation and melt. The two

higher-resolution simulations (3km and 1km) capture the magnitude of peak SWE very accurately, with only a 3 to 6% low bias for 1km and 3km simulations over the accumulation period, respectively. This demonstrates that both simulations may have value for providing accurate estimates of average SWE accumulation across the region.





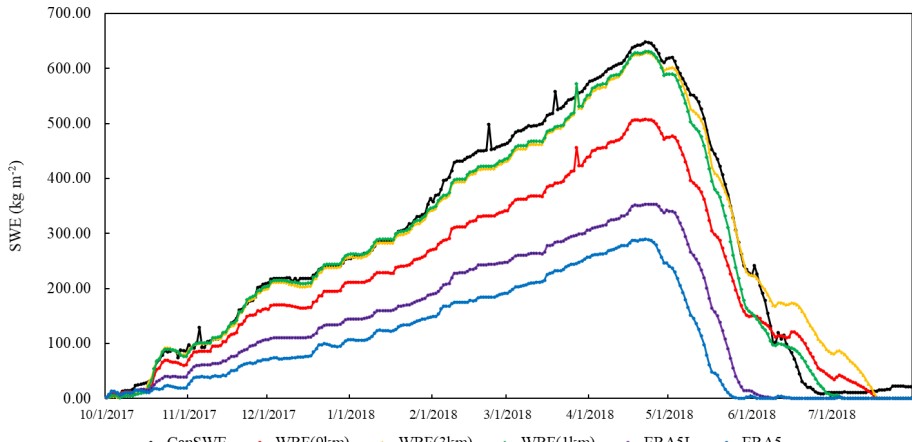

**Figure 2. Temporal variation of SWE, from CanSWE data and WRF model resolution over the SSRB region. The SWE**
**data are aggregated for all stations inside the innermost domain. The spikes in the graph correspond to snowfall events.**

In contrast, the lowest resolution WRF simulation (9km) displays a systematic low bias of about 108 mm (31%) in SWE throughout the accumulation period, suggesting that either there is too little total precipitation reaching the surface at this resolution, or a temperature bias is causing a lower proportion of precipitation to fall as snow. Examining the role of these two forcings on simulated SWE at the three resolutions, we find very close agreement in temperature (Fig.3a), but a systematic low bias in accumulated precipitation at 9km (Fig.3b), indicating that lower total precipitation is the most likely cause of the SWE bias at the lowest resolution. The WRF simulations are configured using a 3-hourly ERA5 forcing at the lateral boundary (i.e., the boundary of the 9km domain). Therefore, the fact that the 9km simulation produces lower total accumulated precipitation than the two higher-resolution simulations over a mountainous region strongly suggests that the cause is orographic enhancement of precipitation within WRF. Interestingly, given that all three of our domains have higher resolutions than ERA5 itself (27 km), this implies that underestimated orographic enhancement may be contributing to a low bias in precipitation at high elevations in ERA5, which in turn leads to a low bias in SWE (Fig.2).

Figure 2 also shows that the WRF simulation at all three resolutions estimates the melting period in two phases: a rapid phase from late April to early June, then a more gradual phase until late June. Also, the difference in melt rate between the two phases is most apparent at the lower resolutions, indicating that melt processes may be more accurately represented at higher resolution based on the melting rate.





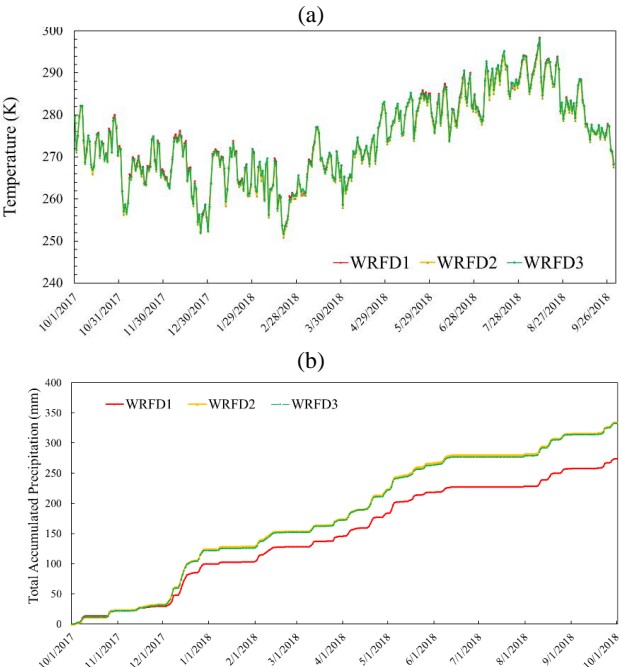

**Figure 3. Temporal variation of 2m temperature (a) and precipitation (b) from the three WRF model resolutions over the SSRB region for the innermost domain.**


Summary statistics for the melting and accumulation period are shown in Figure 4 which presents the RMSE and MB over the region. Generally, Fig. 4 suggests that there is an obvious tendency for RMSE and MB to decrease at finer resolutions. Following ERA5 with 27 km and ERA5-L with 9km resolution, the coarsest model run shows a high value for RMSE especially during accumulation period. WRF-9km underestimate SWE values more than the other two finer

resolutions during both understudied periods, perhaps due to the incapability of the WRF-9km to simulate the processes that are responsible for snow deposition and erosion in mountainous areas that are characterized by heterogeneous snow distribution.

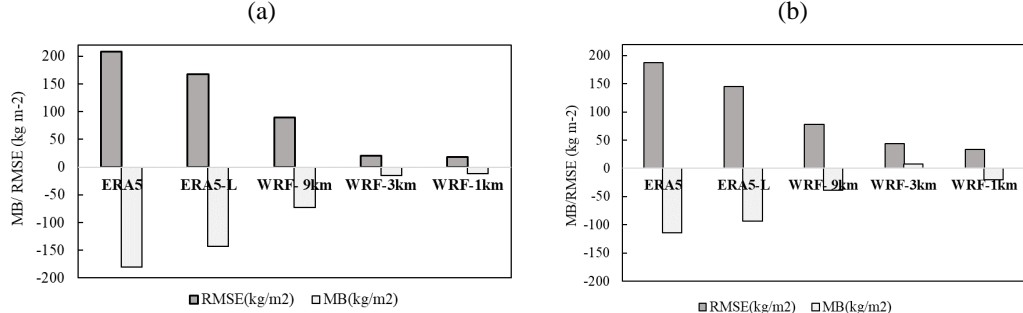

**Figure 4. SWE evaluation during (a) accumulation period (1st October 2017- 22nd April 2018) and (b) melting period (23nd April 2018 to 1st October 2018) over the inner WRF model domain using RMSE and mean bias.**





### 3.2 Evaluation of spatially varying SWE

In this section, the representation of spatial heterogeneity of SWE in the WRF simulations is evaluated by comparing observed and simulated SWE at individual stations within the inner domain. To evaluate the SWE spatial variability, timeseries in each individual station, are depicted in Figure 5. Moreover, the elevation in each station as well as the estimated elevation by WRF simulation at all three resolutions is summarized in Table 2. Mostly in the stations on the leeward side of the mountains, including the four southern stations, there is an underestimation of SWE for all runs. In most stations, the WRF run with the finest resolution shows the best performance. SWE experiences significant changes in both space and time; therefore, the accumulation and the snowpack melting are variable because of the complex topography. During the accumulation period, the difference is more pronounced at each station.

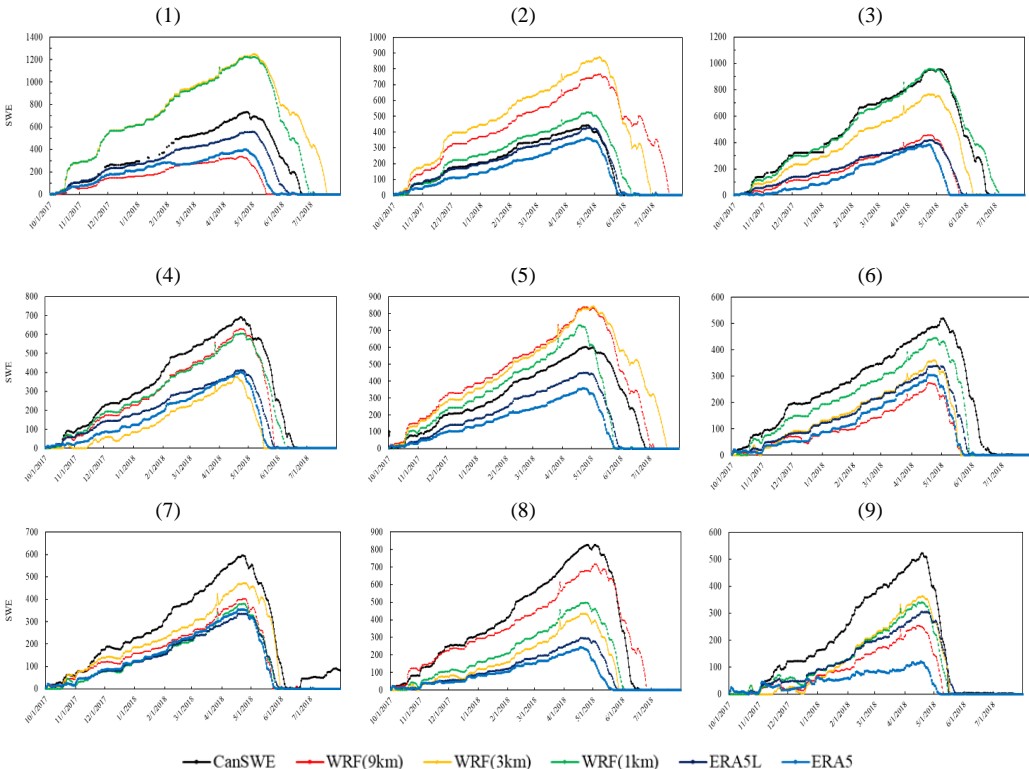

**Figure 5. Estimation of SWE using WRF comparing to the CanSWE data for station 1 to 9.**

**Table 2. Observed elevation of each station vs. the estimated elevation by WRF.**

| | Elevation(m) | | |
|---|---|---|---|
| Station Number | Observation | WRF-9km | WRF-3km | WRF-1km |
| 1 | 2122 | 2099 | 2372 | 2413 |
| 2 | 2120 | 2500 | 2712 | 2225 |
| 3 | 2090 | 2086 | 2233 | 2218 |
| 4 | 2230 | 2199 | 2028 | 2164 |
| 5 | 2160 | 2365 | 2479 | 2483 |
| 6 | 2120 | 2144 | 2163 | 2306 |
| 7 | 2060 | 2242 | 2345 | 2118 |
| 8 | 2130 | 2265 | 2156 | 2158 |
| 9 | 1920 | 1834 | 1894 | 1856 |



Spatial distribution and extent of the SWE for each WRF horizontal resolution are shown in Figure 6(a-f) for accumulation and melting period. As shown in section 3-1, snow accumulates in the mountains from October through April, and snowmelt usually begins in May. Winds may cause redistribution of snow in the alpine zone, with scouring on the windward side of ridges, and deposition on the leeward side (Clow et al., 2012). Both periods show similar spatial distribution of SWE, however, the impact of resolution is obvious. There is a maximum in SWE value in all

three simulations over northern parts of the domain both for accumulation and melting period. During the accumulation period, WRF-1km has larger SWE values in compare with WRF-9km in most areas.

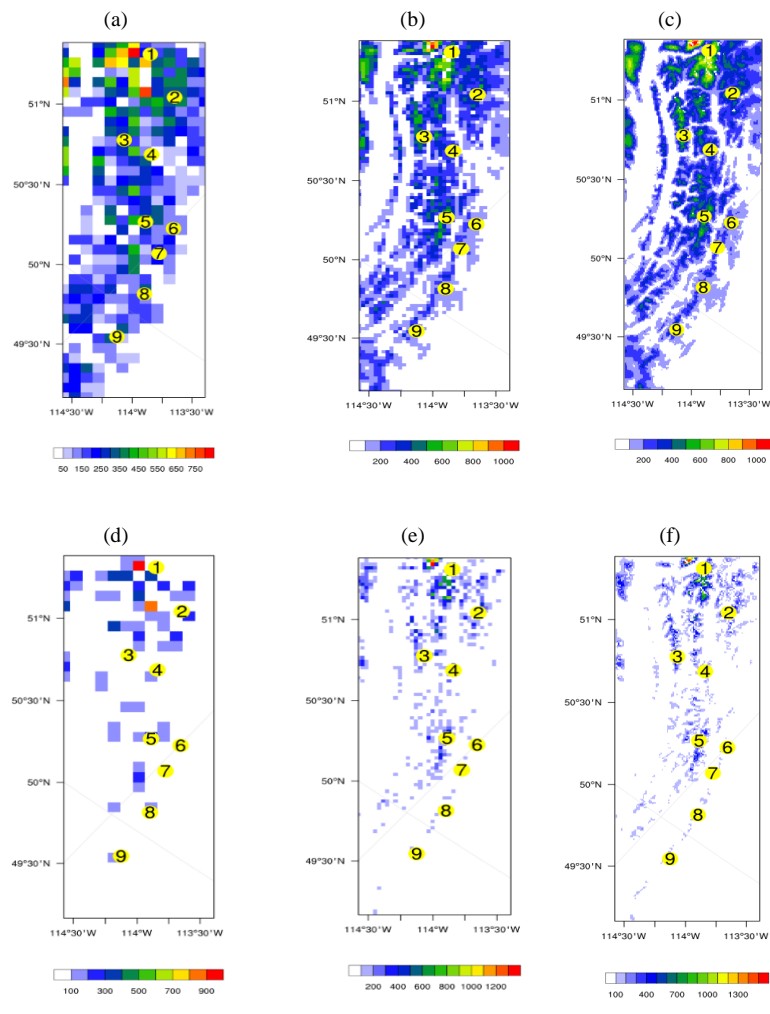

**Figure 6. Simulated mean SWE (kg/m2) from WRF during the accumulation (1st October 2017- 22nd April 2018) and melting (23nd April 2018 to 1st October 2018) period for the 9km(a,d), 3km(b,e) and 1km (c,f) resolution over the inner model domain.**






To show the errors of the estimates spatially, bias and RMSE in each station are compared in Figure 7 for the accumulation and melting periods. This indicates a geographic sensitivity to both bias and RMSE. The RMS errors show that the estimates are more accurate in the southern portion of the domain, where there is also a negative bias. Overall, in most stations, WRF-1km SWE estimation is more reliable than the other two domains, although the difference in some stations is not significant. In the following section, results are analyzed to investigate the role of the terrain elevation in explaining the model errors and biases in SWE simulation.


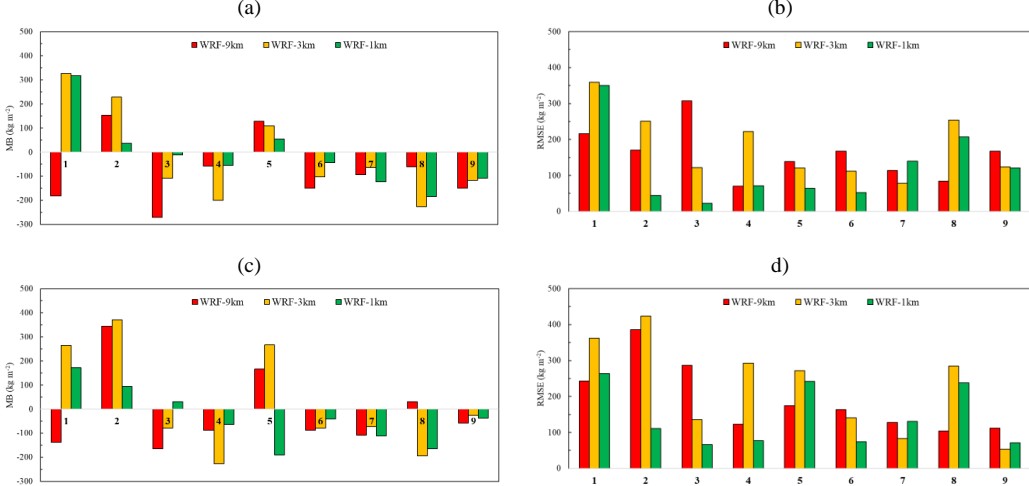

**Figure 7. Evaluation metrics for each station including MB (a,c) and RMSE (b,d) for accumulation (upper row) and melting (lower row) period.**


To study the ability of WRF model to estimate the date and the value of peak SWE at each station, Table 3 and Figure 8 evaluated the estimated peak SWE date and values, respectively. Late April to early May 2018 is an approximate estimation of the peak SWE date over the region in according to the observation (Table 3). Across the nine stations there is an approximately two-week spread in the observed dates of peak SWE between 18 April and 3 May, and the magnitude of the peak appears unrelated to its date. WRF-1km is in good agreement with CanSWE for the date and value of peak SWE in most stations.


**Table 3. Comparison between the date of maximum SWE in CanSWE and anomalies in estimated peak SWE date by WRF in each station during 1st October 2017-2018. The anomaly shows the number of days that peak SWE is before/after CanSWE.**

| Station | Peak SWE Date | Peak SWE Date Anomaly (Day) | | |
|---|---|---|---|---|
| | CanSWE | WRF-9km | WRF-3km | WRF-1km |
| 1 | 24-Apr | -6 | 8 | 8 |
| 2 | 23-Apr | 12 | 12 | -1 |
| 3 | 25-Apr | -3 | -3 | -3 |
| 4 | 23-Apr | -1 | -5 | -1 |
| 5 | 25-Apr | -3 | 7 | -7 |
| 6 | 2-May | -14 | -9 | -9 |
| 7 | 22-Apr | 1 | 0 | 0 |
| 8 | 3-May | -1 | -15 | -11 |
| 9 | 18-Apr | -4 | 0 | 0 |




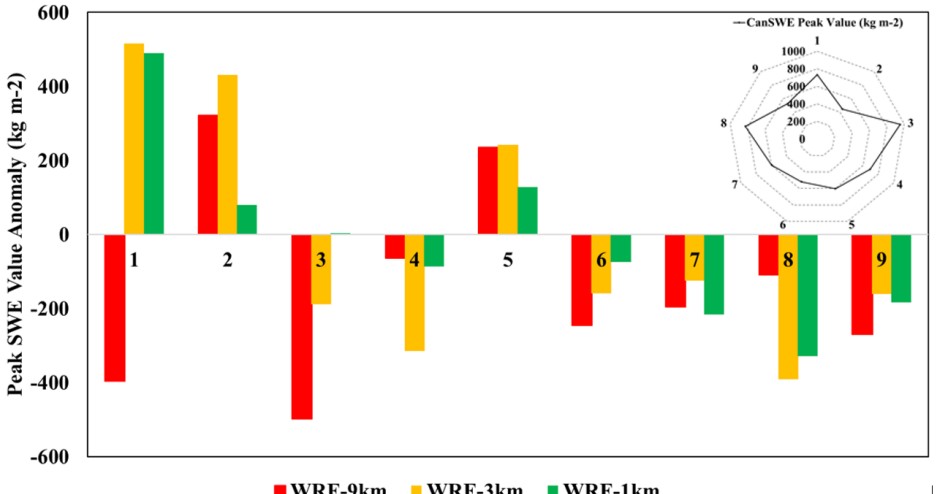

**Figure 8.** **The anomalies in the value WRF maximum SWE estimation for each station during 1st October 2017-2018. The anomaly shows how much the value of SWE is estimated more/less at each resolution (1 to 9). The value of maximum SWE for CanSWE at each station is shown in the upper right.**


Although the timing of the peak SWE is quite similar across all stations, there are differences in the estimated magnitude of maximum SWE in each WRF resolution (Figure 8). In most stations, WRF underestimate the value of the peak SWE, however, it tends to overestimate the value of the peak SWE in the two northern stations.

**3.3 The role of elevation**

There are clear differences in the surface terrain height imposed as the lower boundary condition for the three resolutions of WRF (Fig.9), which suggests a possible role for elevation in the errors in simulated SWE. As is common, the spatial variability in elevation is more pronounced in the highest resolution simulation (Fig.9c) and becomes smoother with decreasing the spatial resolution (Figs.9a,b). In the northern portion of the domain, where the simulated SWE estimates are less accurate, the elevation is higher and more variable than the southern areas. A potential

explanation for the SWE biases is that the 9-km resolution of WRF was not fine enough to resolve the localized peaks in the mountainous topography that experience a cooler mean climate and, typically, higher mean SWE.

Correlation analysis shows that the absolute grid cell elevation is not correlated with SWE MB or RMSE (not shown); however, the elevation bias—the difference between the actual station's elevation and the estimated elevation by the model in a grid cell—does appear to play a significant role. Elevation bias shows a strong positive correlation with MB

at all resolutions (Fig.10a), but an important finding is that the correlation becomes weaker at higher resolutions. In other words, when the elevation biases are largest (at 9km) they are a better indicator of the bias in SWE, whereas at 1km the elevation biases are much smaller and therefore they do not contribute as much to SWE errors. Therefore, it can be deduced that all WRF estimations include uncertainties and biases in simulating SWE, and one important source is biases in the grid cell elevation. We note that elevation bias is not correlated with the RMSE in simulated SWE at any

resolution (Fig.10b); however, correlation coefficient becomes stronger at higher resolutions.





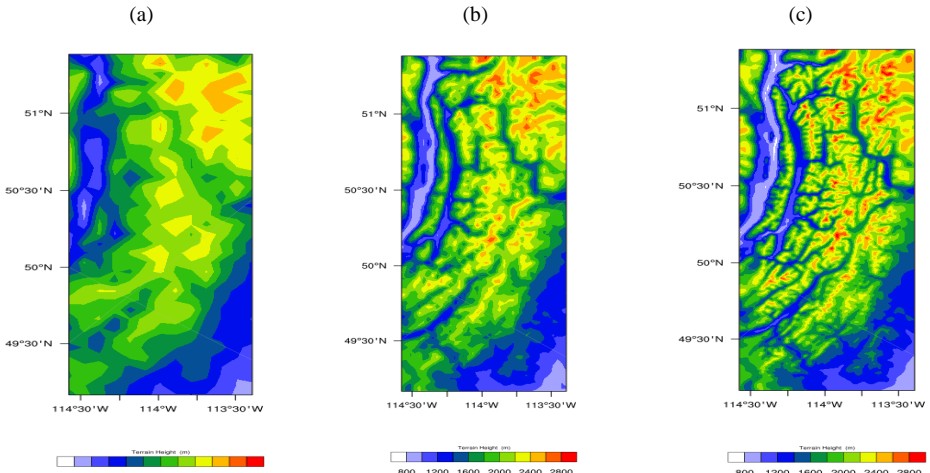

**Figure 9.** Spatial distribution of terrain height over WRF-9km (a), WRF-3km (b) and WRF-1km(c) simulation during the 2018 water year.

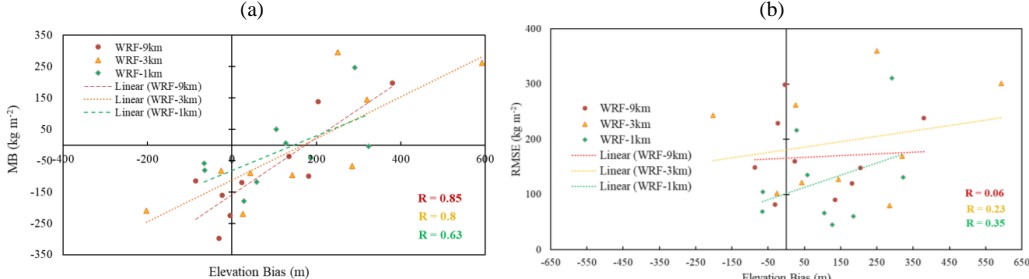

**Figure 10.** Elevation bias against MB (a) and RMSE (b) of SWE estimation.


## 4. Discussion and Conclusions

The objective of this study is to evaluate the potential of the fine horizontal resolution Weather Research and Forecasting (WRF) model to detect the daily values of snow water equivalent (SWE) over the South Saskatchewan River Basin (SSRB) in Western Canada. Three nested domains with fine horizontal resolution of 9, 3 and 1 km are

used. Canadian historical Snow Water Equivalent (CanSWE) dataset is used to evaluate the potential of WRF to detect the spatio-temporal variability in SWE. The evaluation was conducted from 1st October 2017 to 1st October 2018, as the 2018 water year, with average SWE values during 1984 to 2021. Although it is acknowledged that the use of point data for the evaluation of WRF gridded SWE is problematic because of scaling issues, we attempt to mitigate these issues by using a spatial mean taken over all stations. However, earlier studies also showed that the small-scale SWE

can be representative of the grid mean value (e.g. Pan et al., 2003) for the local SWE evaluation.

In general, our initial results over the averaged area, show that all WRF runs behaves nearly similar and show high value of correlation with CanSWE data, though there is a slight difference exist between accumulation and melting period. All WRF estimations mainly tend to underestimate SWE over the whole year, with a largest negative bias in the coarsest resolutions. Results show that WRF fine resolution at 3km and 1km, significantly improves the simulations of



SWE during the year over an averaged area. The coarsest run shows less accuracy during accumulation period, which is likely caused by a systematic bias in accumulated precipitation at 9km. The underestimated precipitation over the mountainous regions at coarse resolutions has been shown by Li et al. (2019). The large bias in the coarsest resolution also may be due to the incapability of the WRF-9km to simulate the processes that are responsible for snow deposition and erosion in mountainous areas (Mott et al., 2018; Raparelli et al. 2021). Over the whole region, there is an obvious

tendency for RMSE and MB to decrease at finer resolutions, therefore, decreasing the horizontal grid spacing within WRF, lead to the reliable SWE estimate over the mountainous SSRB. So, it can be concluded that the accuracy of SWE is closely related to the horizontal resolution. Earlier studies also revealed that there is a dependence of snow estimation on NWP model resolution to capture the orographic processes over the Western Canada and US (Pavelsky et al. 2011; Schirmer and Jamieson, 2015; Wrzesien et al. 2015).

Evaluation of the SWE in individual station showed that there is less amount of snow on the windward side of ridges, and snow deposition on the leeward side. Mostly on the leeward side of the mountains, there is an underestimation of SWE for all WRF performances. There is a maximum in SWE value in all three simulations over northern parts of the domain both for accumulation and melting period. Low temperatures and high cyclonic activity over the northern part of the domain may cause long snow duration and high value of SWE (SWIPA, 2011). Local characteristics of each

station, including the terrain and land cover characteristics, as well as the interactions with the local wind, would play a major role in SWE variability over the region.

Investigating the ability of WRF model to estimate the date and the value of peak SWE in each station reveal that there is an approximately two-week spread in the observed dates of peak SWE between late April and early May, in accordance with the observation. Although the timing of the peak SWE is quite similar across all stations, there are

differences in the estimated magnitude of maximum SWE in each WRF resolution. In most stations, the value of the peak SWE is underestimated, which is consistent with the findings of previous studies (e.g., Jin and Wen, 2012; Wrzesien et al., 2018; He et al., 2019), however, WRF-1km is in good agreement with CanSWE for the date and value of peak SWE. Overall, WRF can also provide reliable data for peak SWE date and value, especially in fine horizontal resolution.

Analysis of the role of elevation shows that elevation itself doesn't show any correlation with MB and RMSE however the elevation bias shows a strong positive correlation with MB at all resolutions, which becomes weaker at higher resolutions and would be a better indicator of the bias in SWE in coarse resolution. This result highlights an important consideration when comparing point observations to output from a model grid cell, namely that the agreement in elevation between any individual station and the model's mean elevation in a grid cell will be closer, on average, for a

1 km grid cell compared to a 9 km grid cell. This is because the actual topography and associated elevation will typically be much more variable over an area of 81 km2 than over 1 km2. Another way to frame this is that an individual station is significantly less representative of the actual variations in elevation, precipitation, SWE etc. within a 9 km grid cell than a 1 km grid cell, and so one might find better agreement between the model and a distributed network of stations within the grid cell. Unresolved topography contributes to the inaccurate SWE estimation in the

coarse resolution. The bias in elevation, meaning the lower or upper mountains, affect the condensation of water vapor, precipitation, topography related temperature and therefore the amount of estimated SWE over the region. An earlier study also showed that increasing resolution in regional models resolve more small-scale features (Xu et al., 2019) and therefore improve SWE estimation. On the other hand, as the elevation bias does not impact the RMSE in any



resolution, RMSE likely depends on other atmospheric variables rather than elevation. Therefore, additional perspectives could be considered in future work to better understand the mechanisms and potential cause of uncertainties in SWE estimations over the mountains.

In the end, this study has shown that high resolution WRF can provide reliable and reasonable estimates of SWE values as an input data for accurate hydrologic modeling, required for runoff forecasts. Analysis presented in this paper revealed that WRF's high resolution can represent spatio-temporal variability of SWE over the mountainous

region, and it is expected to be helpful for flood forecasting in mountainous regions. However, further work is needed to remove the biases and capture the accurate value of SWE over the western Canada.

**Acknowledgments**

We greatly acknowledge Neha Kanda from the University of Waterloo for her technical support in this work.

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
