# Peer review of "The importance of model horizontal resolution for improved estimation of Snow Water Equivalent in a mountainous region of Western Canada"

_EGUsphere, 2024_

## Referee Comment (RC1)

A review of: "*The importance of model horizontal resolution for improved estimation of Snow Water Equivalent in a mountainous region of Western Canada*" by Samaneh Sabetghadam Christopher G. Fletcher and Andre Erler

Brief Summary: In this paper, the authors attempt to show the importance of high-resolution climate models for estimating SWE in mountainous regions. The study basin has 9 available CanSWE snow pillows with a continuous record. Comparisons of model SWE vs CanSWE records are made for 5 modeled datasets: WRF 9/3/1km, ERA5 and ERA5-Land. The authors chose to aggregate the 9 stations for the bulk of the analysis, while the individual stations are investigated in later sections. A brief investigation into the role of elevation and elevation bias in the models are made as well. The authors conclude that WRF at 1/3km resolution can provide a good estimate of SWE values in the South Saskatchewan River Basin and thus can be used for flood/runoff forecasting.

Overall Thoughts: The paper is well written and easy to understand. I have minor comments for some of the language used. I have two major concerns that I believe the authors should address prior to publication: I believe not enough snow pillow sites were used in the analysis, and the comparison of SWE point data to gridded model SWE is not justified adequately. I have included references of papers that expand upon these major concerns.

Major Comments:
1. This study uses only 9 snow pillow sites for an area of ~ 100km x 60km. I have concerns that the number of stations used is too low. The authors do not present any work to suggest that this number of stations is statistically significant and thus viable to compare against the gridded data set. With a sample size of only 9 stations, the aggregation of the stations to compare against the modeled SWE does not seem appropriate. Further work to show that these results are statistically significant and unbiased.
   a. In line with this comment, I suggest the authors include standard deviation and/or mean absolute error metrics for their results.
2. The SWE time series shown in figure 2 are based off the spatially aggregated station results. However, when you look at the SWE time series for the individual stations (figure 5), the results seem at odds with figure 2. There are several instances of the "best" data product (WRF 1k) differing from CanSWE by what appears to be 50% or more. Therefore, I suspect the aggregate results is likely averaging out the mischaracterized stations. This discrepancy is glossed over in the manuscript and takes away from any claims the authors attempt to make about the relative value of high-resolution WRF runs. Again, calculation standard deviation and mean absolute bias/error for the individual stations and expanding the discussion of those results would be helpful.
3. The comparison of point data SWE to gridded data SWE is difficult *(see Blöschl, 1999)*. By design, snow pillows are not representative of typical mountain terrain. Snow pillows are placed in (relatively) open and flat clearings where they can accurately report on the mass of falling snow while being protected from the wind from surrounding tree cover. However, the processes that impact the actual spatial distribution of snow in mountains

are not well represented by the local environment that is conducive to snow pillow placement. Many of the processes that govern snow depth in mountain environments are happening on sub 1km scales. Preferential deposition and vegetation presence are important factors that operate on the <100m scale, which is not resolved by any of the model resolutions in this study. See Clark et al., (2011) for more details on these processes.

The authors claim the higher resolution modeled runs are more accurate, but the results presented do not show this. They show that the spatial variability of the grid cell sizing is impacting the SWE estimate from the model. This can be clearly seen in the elevation bias component of the results section (specifically figure 10). Looking at Station 1, the elevation bias for the 3km and 1km model runs are ~+300m and ~+260m respectively. This results in a massive overestimation of snow accumulation by the model. Meanwhile, the 9km model run has an elevation bias of only -30m and shows the typical WRF underestimation of SWE (i.e. Wrzesien et al., 2018)

   a. How can the authors be sure their results are showing the differences in model resolution as opposed to the difference in spatial variability due to the elevation bias in the model?
   b. See *Scaling Issues in Snow Hydrology* (Blöschl, 1999)for more details.
   c. See Rice & Bales (2010)and Dressler et al., (2006) for more info on snow pillow site selection.

Minor Comments:

1. The authors do not refer to the model simulations with consistent language. For example, figure 3 refers to them as WRFD1/D2/D3. While the plot colors match figure 2 which refers to them explicitly as WRF(9km)(3km)(1km), the reader is forced to double check against the figure. I suggest the authors define a consistent naming format early in the paper and stick to it. It could be as simple as "the 9 km WRF runs will be referred to as WRF9K for the remainder of this paper" etc.
2. The authors bounce around between their use of increase/decrease, fine/coarse, high/low in a confusing manner. A more consistent verbiage would make it easier for the reader. Improving the precision of language would increase the readability of the paper. For example, phrases like "less than 4km resolution" are very confusing.
   a. Interpreted literally, one could assume they meant "a resolution with a numeric value lower than 4". i.e., a finer resolution
   b. However, it is equally fair to interpret that as "a resolution with less spatial fidelity than 4km" i.e., a coarser resolution.
3. The introduction reads poorly. The layout does not flow well and feels disjointed in some of the paragraph transitions. Please consider restructuring.
   a. Line 97 has an extra "ERA5 and ERA5-land" in it.
4. The hypothesis of this entire manuscript feels unclear to me. I'm unsure if the authors believe WRF estimates of SWE are appropriate for use at basin scale, grid scale, or both.

      a. My inclination is that they believe it is appropriate for basin scale estimation, but discussion around the analysis of individual stations also leads me to believe they are implying some level of reliability in estimating SWE at grid/point scale.

      b. I highly suggest the authors refine the scope and goals of this manuscript and reframe the title/hypothesis/abstract/conclusion appropriately.

5. I feel that the inclusion of ERA5 and ERA5-L in the analysis is mostly left out. In the current state it feels like it has been added just to have an extra point of comparison. I feel that the authors should expand upon this analysis in a written discussion or remove it.

6. The investigation in elevation bias in section 3.3 is interesting but not fully explored. I suggest the authors expand on this section.

      a. Line 301 states "when elevation biases are largest (at 9km)" but both figure 10 and table 2 show that the largest bias (both positive and negative) occur in the 3km runs. Please clarify if this is an error or if some other metric (possibly mean absolute bias?) was used to justify this statement.

In conclusion, the work presented in this manuscript has significant issues in the underlying assumptions made by the authors. Should the authors reframe their hypothesis and conclusions based on their results, a further iteration of this work could be publishable. The defense of the statistical significance of this work with only 9 snow pillow sites is required. Changes in their justification of their comparison of snow-pillow SWE to gridded model results is necessary. A significantly expanded discussion of the individual station analysis is necessary as well. Further statistical error metrics are necessary for those individual station results. Finally, I recommend the authors expand upon the elevation bias section of the manuscript.

***References:***

Blöschl, G. (1999). Scaling issues in snow hydrology. *Hydrological Processes*, *13*(14–15), 2149–2175. https://doi.org/10.1002/(SICI)1099-1085(199910)13:14/15<2149::AID-HYP847>3.0.CO;2-8

Clark, M. P., Hendrikx, J., Slater, A. G., Kavetski, D., Anderson, B., Cullen, N. J., Kerr, T., Örn Hreinsson, E., & Woods, R. A. (2011). Representing spatial variability of snow water equivalent in hydrologic and land-surface models: A review. *Water Resources Research*, *47*(7), 2011WR010745. https://doi.org/10.1029/2011WR010745

Dressler, K. A., Fassnacht, S. R., & Bales, R. C. (2006). A Comparison of Snow Telemetry and Snow Course Measurements in the Colorado River Basin. *JOURNAL OF HYDROMETEOROLOGY*, *7*.

Rice, R., & Bales, R. C. (2010). Embedded-sensor network design for snow cover measurements around snow pillow and snow course sites in the Sierra Nevada of California. *Water Resources Research*, *46*(3). https://doi.org/10.1029/2008WR007318 open_in_new

Wrzesien, M. L., Durand, M. T., Pavelsky, T. M., Kapnick, S. B., Zhang, Y., Guo, J., & Shum, C. K. (2018). A New Estimate of North American Mountain Snow Accumulation From Regional Climate Model Simulations. *Geophysical Research Letters*, *45*(3), 1423–1432. https://doi.org/10.1002/2017GL076664

---

## Author Comment (AC1)

A review of: *"The importance of model horizontal resolu4on for improved estimation of SnowWater Equivalent in a mountainous region of Western Canada"* by Samaneh Sabetghadam Christopher G. Fletcher and Andre Erler

Brief Summary: In this paper, the authors attempt to show the importance of high-resolution climate models for estimating SWE in mountainous regions. The study basin has 9 available CanSWE snow pillows with a continuous record. Comparisons of model SWE vs CanSWE records are made for 5 modeled datasets: WRF 9/3/1km, ERA5 and ERA5-Land. The authors chose to aggregate the 9 stations for the bulk of the analysis, while the individual stations are investigated in later sections. A brief investigation into the role of elevation and elevation bias in the models are made as well. The authors conclude that WRF at 1/3km resolution can provide a good estimate of SWE values in the South Saskatchewan River Basin and thus can be used for flood/runoff forecasting.

Overall Thoughts: The paper is well written and easy to understand. I have minor comments for some of the language used. I have two major concerns that I believe the authors should address prior to publication: I believe not enough snow pillow sites were used in the analysis, and the comparison of SWE point data to gridded model SWE is not justified adequately. I have included references of papers that expand upon these major concerns.

**Response**: The authors would like to thank the reviewer for the time and valuable comments and suggestions. Below the reviewer comments are in blue, our responses are in black. All changes are highlighted in red in the revised manuscript.

**Major Comments:**

1. This study uses only 9 snow pillow sites for an area of ~ 100km x 60km. I have concerns that the number of stations used is too low. The authors do not present any work to suggest that this number of stations is statistically significant and thus viable to compare against the gridded data set. With a sample size of only 9 stations, the aggregation of the stations to compare against the modeled SWE does not seem appropriate. Further work to show that these results are statistically significant and unbiased.
a. In line with this comment, I suggest the authors include standard deviation and/or mean absolute error metrics for their results.

**Response**: We do agree that having more data can lead to more accurate and reliable results, however, it's also important to consider the quality of the data. The presence of a small amount of high-quality station providing daily SWE data over the study period, can lead to meaningful insights. Following an extensive investigation of the CANSWE dataset, we identified nine stations meeting this criterion, meaning they consistently offer daily SWE values with minimal data gaps. These stations have been selected based on their consistent daily SWE data availability among stations in the region. Though, based on the reviewer's suggestion, we also add mean absolute error and standard deviation metrics to our statistical analysis. Figure 1, 4, 7 and 10 has been updated. More analysis is added to the manuscript about the agreement between the metrics which confirms that the results are unbiased and statistically significant.

2. The SWE time series shown in figure 2 are based off the spatially aggregated station results. However, when you look at the SWE time series for the individual stations (figure 5), the results seem at odds with figure 2. There are several instances of the "best" data product (WRF 1k) differing from CanSWE by what appears to be 50% or more. Therefore, I suspect the aggregate results is likely

averaging out the mischaracterized stations. This discrepancy is glossed over in the manuscript and takes away from any claims the authors attempt to make about the relative value of high resolution WRF runs. Again, calculation standard deviation and mean absolute bias/error for the individual stations and expanding the discussion of those results would be helpful.

**Response**: Based on the reviewer's comment, mean absolute error and standard deviation are calculated for individual stations as well. Discussion is added to the text about the agreement between the metrics. As mentioned in the manuscript, aggregation of the station may smooth the differences between estimations and observation. This effect is introduced in the literature as aggregation filtering. Additionally, the point-to-grid comparison can introduce uncertainties in the verification results for individual stations. This has been added to the text for more clarification. Also, it should be noted that, our primary focus is on estimating spatio-temporal variation of SWE across the SSRB region, rather than claiming WRF's precision in estimating SWE at a specific point scale.

3. The comparison of point data SWE to gridded data SWE is difficult *(see Blöschl, 1999)*. By design, snow pillows are not representative of typical mountain terrain. Snow pillows are placed in (relatively) open and flat clearings where they can accurately report on the mass of falling snow while being protected from the wind from surrounding tree cover. However, the processes that impact the actual spatial distribution of snow in mountains are not well represented by the local environment that is conducive to snow pillow placement. Many of the processes that govern snow depth in mountain environments are happening on sub 1km scales. Preferential deposition and vegetation presence are important factors that operate on the <100m scale, which is not resolved by any of the model resolutions in this study. See Clark et al., (2011) for more details on these processes.
The authors claim the higher resolution modeled runs are more accurate, but the results presented do not show this. They show that the spatial variability of the grid cell sizing is impacting the SWE estimate from the model. This can be clearly seen in the elevation bias component of the results section (specifically figure 10). Looking at Station 1, the elevation bias for the 3km and 1km model runs are ~+300m and ~+260m respectively. This results in a massive overestimation of snow accumulation by the model. Meanwhile, the 9km model run has an elevation bias of only -30m and shows the typical WRF underestimation of SWE (i.e. Wrzesien et al., 2018)
a. How can the authors be sure their results are showing the differences in model resolution as opposed to the difference in spatial variability due to the elevation bias in the model?
b. See *Scaling Issues in Snow Hydrology* (Blöschl, 1999) for more details.
c. See Rice & Bales (2010) and Dressler et al., (2006) for more info on snow pillow site selection.

**Response**: Thanks for pointing out this issue. As stated in the text, we acknowledged that the point comparison is challenging. We add discussions to the result section based on the suggested references. All suggested references have been read in detail and referred in the manuscript. We mentioned that elevation bias is not significantly correlated with error metrics at any resolution. On average, the elevation bias has the highest value in WRF3K, but the SWE has better performance in WRF1K. We supposed that estimated errors in SWE likely depends on other atmospheric variables rather than elevation and additional perspectives could be considered in future work to better understand the mechanisms and potential cause of uncertainties in SWE estimations over the mountains. This needs accurate evaluation of elevation estimates by WRF over the mountainous regions and would be the focal point of our next study.

**Minor Comments:**

1. The authors do not refer to the model simulations with consistent language. For example, figure 3 refers to them as WRFD1/D2/D3. While the plot colors match figure 2 which refers to them explicitly as WRF(9km) (3km) (1km), the reader is forced to double check against the figure. I suggest the authors define a consistent naming format early in the paper and stick to it. It could be as simple as "the 9 km WRF runs will be referred to as WRF9K for the remainder of this paper" etc.

**Response**: It has been changed to a consistent naming format thorough the main manuscript, figures and tables. WRF9K, WRF3K and WRF1K referred to 9, 3 and 1 km WRF runs respectively. Those are introduced in the section 2.

2. The authors bounce around between their use of increase/decrease, fine/coarse, high/low in a confusing manner. A more consistent verbiage would make it easier for the reader. Improving the precision of language would increase the readability of the paper. For example, phrases like "less than 4km resolution" are very confusing.
a. Interpreted literally; one could assume they meant "a resolution with a numeric value lower than 4". i.e., a finer resolution
b. However, it is equally fair to interpret that as "a resolution with less spatial fidelity than 4km" i.e., a coarser resolution.

**Response**: The manuscript has been revised and more consistent language has been replaced where it is confusing.

3. The introduction reads poorly. The layout does not flow well and feels disjointed in some of the paragraph transitions. Please consider restructuring.

**Response**: Following this suggestion it has been restructured and revised.

a. Line 97 has an extra "ERA5 and ERA5-land" in it.

**Response**: It has been removed.

4. The hypothesis of this entire manuscript feels unclear to me. I'm unsure if the authors believe WRF estimates of SWE are appropriate for use at basin scale, grid scale, or both.
a. My inclination is that they believe it is appropriate for basin scale estimation, but discussion around the analysis of individual stations also leads me to believe they are implying some level of reliability in estimating SWE at grid/point scale.
b. I highly suggest the authors refine the scope and goals of this manuscript and reframe the title/hypothesis/abstract/conclusion appropriately.

**Response**: There are few studies have been performed regionally (especially in Canada) to validate the WRF model estimation of the spatial and temporal patterns in SWE and other snow properties. Previous studies (mostly over the USA) shows that model horizontal resolution is one of the key factors that should be improved to increase the accuracy of a simulated snowpack. The main goal of the current paper is to evaluate the potential of high-resolution WRF model run to correctly simulate the daily values of snow water equivalent over the SSRB region in Canada. Results indicate that WRF performs well in estimating SWE over an entire watershed. However, it is important to note that we do not assert that WRF's accuracy in estimating SWE at a specific point scale. This clarification has been included in the text.

5. I feel that the inclusion of ERA5 and ERA5-L in the analysis is mostly left out. In the current state it feels like it has been added just to have an extra point of comparison. I feel that the authors should expand upon this analysis in a written discussion or remove it.

**Response**: ERA5 and ERA5-L are used to evaluate their SWE values (as the initial conditions for WRF) and to understand the role of resolution in SWE estimation. ERA5-L is widely regarded as the state-of-the-art land surface reanalysis and offers a relatively high resolution (9km), homogenous, long-term observation-based data record. As a gridded, model-based product, ERA5L provides the ideal benchmark against which other observation-based model estimates should be compared. Our WRF simulations are driven by ERA5 atmospheric forcing at the boundary. Therefore, we wish to assess whether the WRF simulations perform better than ERA5L at estimating aggregated SWE over SSRB. The ERA5 results suggest that improved resolution improves SWE estimation. It's interesting that the ERA5L SWE (9km) performs less well than the WRF9K simulation. Our preliminary results (from Fig.2, 4 and 5) show that ERA5 and ERA5-L underestimate SWE values and there is a distinct difference between SWE from these reanalysis data and observational data. So, for the rest of the evaluation, we focused on the WRF estimation.

6. The investigation in elevation bias in section 3.3 is interesting but not fully explored. I suggest the authors expand on this section.
a. Line 301 states "when elevation biases are largest (at 9km)" but both figure 10 and table 2 show that the largest bias (both positive and negative) occur in the 3km runs. Please clarify if this is an error or if some other metric (possibly mean absolute bias?) was used to justify this statement.

**Response**: Thank you very much for highlighting this issue, this is an error and it has been corrected. As previously noted, the elevation bias is typically highest in WRF3K, while SWE performance is the best in WRF1K. Then we hypothesize that errors in SWE estimation maybe influenced by variables beyond elevation. This has been added to the manuscript.

In conclusion, the work presented in this manuscript has significant issues in the underlying assumptions made by the authors. Should the authors reframe their hypothesis and conclusions based on their results, a further iteration of this work could be publishable. The defense of the statistical significance of this work with only 9 snow pillow sites is required. Changes in their justification of their comparison of snow-pillow SWE to gridded model results is necessary. A significantly expanded discussion of the individual station analysis is necessary as well. Further statistical error metrics are necessary for those individual station results. Finally, I recommend the authors expand upon the elevation bias section of the manuscript.

**Response**: All suggested changings have been made accordingly to the reviewer's comment which improves the quality of the paper.

[revised manuscript text omitted]

---

## Author Comment (AC2)

The authors would like to thank the reviewer for the time and valuable comments and suggestions. Below the reviewer comments are in blue, our responses are in black. All changes are highlighted in red in the revised manuscript.

- In general it is hard to read the figures — the authors should consider saving them at a higher resolution and/or increasing font sizes. This is making it harder to interpret the figures.

**Response**: Figure 2, 4, 5, 6, 7, 8 and 10 have been updated. Font sizes and resolution have been enhanced.

- "Figure 2. Temporal variation of SWE, from CanSWE data and WRF model resolution over the SSRB region. The SWE data are aggregated for all stations inside the innermost domain. The spikes in the graph correspond to snowfall events."

  Do the spikes imply that the snow accumulated and then melts, or that this is a data artifact? The authors should state what they think is actually causing the spikes. It seems unusual to have the spikes consistent across all of the CanSWE observations.

**Response:** The spikes are due to the snowfall events, indicating the accumulation and subsequent melting of snow. This information has also been included in the manuscript for further clarification.

- "Figure 6. Simulated mean SWE (kg/m2)"

  $kg/m^2$ should be properly formatted

**Response**: It has been corrected**.**

"is is because the actual topography and associated elevation will typically be much more variable over an area of 81 km2 than over 1 km..."

Same here — correctly format km2.

**Response:** It has been corrected**.**

"The large bias in the coarsest resolution also may be due to the incapability of the WRF-9km to simulate the processes that are responsible for snow deposition and erosion in mountainous areas (Mott et al., 2018; Raparelli et al. 2021)."

It is not clear what "snow erosion" is in this context. Maybe "snow redistribution"? The latter is the more commonly used term.

**Response:** It has been replaced by "snow redistribution".

[revised manuscript text omitted]

---

## Referee Report (RR1)

Throughout the manuscript you use SD for Standard Deviation. Many snow papers use SD for snow depth, consider an alternative abbreviation (StDv/ SDv etc) or spelling it out every time for clarity! I found this confusing and had to reread multiple times when I thought the manuscript was referring to snow depth.

Line 67 - regional weather forecast models are recently used -> regional weather forecast models have been used recently.
Line 69 - This line is not grammatically correct. Remove significantly.
Line 70 - remove global snow hydrology or explain what this means.
Line 107-08 - "high-resolution Weather Research and Forecasting (WRF) model run" -> Weather and Research Forecasting model run at various resolutions.
Line 112 - "on the accurate estimation of peak SWE" -> rewrite this to be more grammatically correct.
Line 113-114 "This study can provide information related to the regional water management and hazard prevention. " Expand or remove this sentence.
Line 145 - Please rephrase to emphasize these are weather stations with snow pillows.

Line 241 - "The ERA5-L SWE at 9km resolution performs less well than the WRF9K simulation" Less well is not grammatically correct, please fix.

Section 3-1. I am unsure what "SD - CanSWE SD" means. Is this the SD of each dataset minus the SD of the CANSWE dataset? If so, can you explain this in the text more directly. Also, if so, can you give a better justification of why you showed this as opposed to just the standard deviation? I am not sure the Stdev of the mean time series makes sense as opposed to calculating the Stdev of the spatial distribution of SWE? It makes sense that the Stdev of the time series is strange as obviously SWE starts at zero and ends at zero, so we have a large distribution of possible SWE values over the series. Or did you calculate the Stdev for each day and then calculate the SD time series and then a total average over the time series? This section really needs more clarity. Also, if you keep the Stdev of this daily time series, please fix the caption, the last sentence doesn't make sense.

Line 283 - Remove "obvious"
Line 287 - swap the order of "deposition" and "redistribution."

Table 2 - It might be nice to include the average/mean elevation bias for WRF9/3/1k.

Figure 7 - Are you sure that the increased error in the melt season is not due to the incorrect temperature from lapse-rate correction of the elevation bias incorrectly influencing the melt timing?

Table 5 - Clarify what the signs of the numbers mean. I mean have missed it, but I could not find. Is a negative number the peak SWE date is earlier than CanSWE or later?

Line 394-408 - In this section you start by saying that the elevation bias is correlated to mean bias error, but then later on in line 403 you say the "elevation bias is not significantly correlated with error metrics at any resolution" Please clarify this statement and rewrite this paragraph as necessary. Is mean bias not an error metric?

Line 478 - Please rephrase, elevation bias does not affect RMSE MAE or SD, but it does impact MB. This in turn influences your RMSE and MAE Thus, you cannot conclude that elevation bias is not a factor in your error statistics.

---

## Author Response (AR2)

**Response to the Reviewer #1**:

The authors would like to thank the reviewer for the time and valuable comments and suggestions. Below the reviewer comments are in blue, our responses are in black. All changes are highlighted in red in the revised manuscript.

Throughout the manuscript you use SD for Standard Deviation. Many snow papers use SD for snow depth, consider an alternative abbreviation (StDv/ SDv etc) or spelling it out every time for clarity! I found this confusing and had to reread multiple times when I thought the manuscript was referring to snow depth.
**Response**: Thanks for pointing out this issue. SD has been changed to StDv thorough the main manuscript, all figures and tables.

Line 67 - regional weather forecast models are recently used -> regional weather forecast models have been used recently.
**Response:** It has been corrected**.**

Line 69 - This line is not grammatically correct. Remove significantly.
**Response:** It has been removed**.**

Line 70 - remove global snow hydrology or explain what this means.
**Response:** It has been removed**.**

Line 107-08 - "high-resolution Weather Research and Forecasting (WRF) model run" -> Weather and Research Forecasting model run at various resolutions.
**Response:** It has been corrected**.**

Line 112 - "on the accurate estimation of peak SWE" -> rewrite this to be more grammatically correct.
**Response:** It has been corrected as: Particular attention is paid to investigate the role of the WRF model's grid cell size on the accurate estimation of peak SWE timing and value across the watershed.

Line 113-114 "This study can provide information related to the regional water management and hazard prevention. " Expand or remove this sentence.
**Response:** It has been removed**.**

Line 145 - Please rephrase to emphasize these are weather stations with snow pillows.
**Response:** It has been rephrased as: Nine weather stations, equipped with automated snow pillows, have been selected based on the availability of daily SWE data with minimal data gaps during the study period.

Line 241 - "The ERA5-L SWE at 9km resolution performs less well than the WRF9K simulation" Less well is not grammatically correct, please fix.
**Response:** It has been corrected**.**

Section 3-1. I am unsure what "SD - CanSWE SD" means. Is this the SD of each dataset minus the SD of the CANSWE dataset? If so, can you explain this in the text more directly. Also, if so, can you give

a better justification of why you showed this as opposed to just the standard deviation? I am not sure the Stdev of the mean time series makes sense as opposed to calculating the Stdev of the spatial distribution of SWE? It makes sense that the Stdev of the time series is strange as obviously SWE starts at zero and ends at zero, so we have a large distribution of possible SWE values over the series. Or did you calculate the Stdev for each day and then calculate the SD time series and then a total average over the time series? This section really needs more clarity. Also, if you keep the Stdev of this daily time series, please fix the caption, the last sentence doesn't make sense.

**Response:** Yes, "StDv - CanSWE StDv" refers to the standard deviation of each dataset minus the StDv of the CanSWE dataset. The text has been revised to clarify to avoid confusion. The rationale for using this approach was to highlight the relative variability of each dataset in comparison to the CanSWE dataset, which serves as a benchmark. The caption also has been revised.

Line 283 - Remove "obvious"
**Response:** It has been removed**.**

Line 287 - swap the order of "deposition" and "redistribution."
**Response:** It has been changed as suggested**.**

Table 2 - It might be nice to include the average/mean elevation bias for WRF9/3/1k.
**Response:** Average elevation bias (in percent) for WRF simulation at all three resolutions has been added to the table**.**

Figure 7 - Are you sure that the increased error in the melt season is not due to the incorrect temperature from lapse-rate correction of the elevation bias incorrectly influencing the melt timing?
**Response:** It is possible that the increased error during the melt season could be attributed to the incorrect temperature resulting from lapse-rate correction of elevation bias affecting melt timing, however, this aspect was not examined in the current study. However, it is an important factor that could be explored in future research

Table 5 - Clarify what the signs of the numbers mean. I mean have missed it, but I could not find. Is a negative number the peak SWE date is earlier than CanSWE or later?
**Response:** It has been clarified in the caption as: The anomaly shows the number of days that peak SWE occurs before (denoted by a minus sign) and after CanSWE.

Line 394-408 - In this section you start by saying that the elevation bias is correlated to mean bias error, but then later on in line 403 you say the "elevation bias is not significantly correlated with error metrics at any resolution" Please clarify this statement and rewrite this paragraph as necessary. Is mean bias not an error metric?
**Response:** It has been changed to "the other understudied error metrics"

Line 478 - Please rephrase, elevation bias does not affect RMSE MAE or SD, but it does impact MB. This in turn influences your RMSE and MAE Thus, you cannot conclude that elevation bias is not a factor in your error statistics.

**Response:** It has been rephrased for more clarification**.**

**Response to the Reviewer #2:**

The authors would like to thank the reviewer for the time and valuable comments and suggestions. Below the reviewer comments are in blue, our responses are in black. All changes are highlighted in red in the revised manuscript.

The study evaluates the Weather Research and Forecasting (WRF) model at three different resolutions (9km, 3km, 1km) for estimating Snow Water Equivalent (SWE) in Western Canada's South Saskatchewan River Basin (SSRB). It uses CanSWE dataset for validation during the 2017-2018 water year, Employs ERA5 reanalysis data for boundary conditions
Major Findings:
• Higher resolution (1km and 3km) WRF simulations significantly improve SWE estimates compared to coarser resolution (9km)
• The 9km resolution consistently underestimates SWE by about 31%
• Peak SWE timing is well captured across all resolutions, occurring in late April
• Strong correlation between elevation bias and SWE bias, particularly at coarser resolutions
Scientific Merit:
• The methodology is robust, using established models and validation techniques
• Results align with theoretical expectations about resolution impacts on mountainous precipitation
• Good use of statistical analysis to quantify model performance
Areas for Improvement:
From a climate modeling perspective:
• The study could benefit from analyzing more water years to ensure results are not year-specific
• Further investigation of physical mechanisms behind resolution-dependent biases would strengthen the findings
• Discussion of computational costs vs. benefits of higher resolution would be valuable
Certain sections could be rewritten for better clarity, here some examples:

1 - From Abstract:

Original: "All WRF simulations tend to underestimate annual SWE, with largest biases (up to 58 kg/m2, i.e. relatively 24%) found at higher elevations and in simulations at coarser horizontal resolution."

Can be improved: "All WRF simulations underestimated the annual SWE. The largest errors occurred in two conditions: at higher elevations and when using coarser horizontal resolution. These biases reached up to 58 kg/m2 (24% relative error)."

**Response:** It has been changed as suggested.

2 - From Introduction:

Original: "Spatiotemporal distribution of SWE, particularly within northern latitudes and higher elevations, shows the extent of spring and summer runoff (Barnet et al., 2005; King et al., 2020)."

**Response:** It has been changed as suggested.

Can be improved: "The distribution of SWE across space and time is especially important in northern regions and at high elevations. This distribution determines how much water will be available during spring and summer runoff periods (Barnet et al., 2005; King et al., 2020)."

3 - From Results:

Original: "Examining the role of these two forcings on simulated SWE at the three resolutions, we find very close agreement in temperature (Fig.3a), but a systematic low bias in accumulated precipitation at WRF9K (Fig.3b), indicating that lower total precipitation is the most likely cause of the SWE bias at the lowest resolution."

Can be improved: "We examined how temperature and precipitation affect SWE simulation at different resolutions. Temperature values were very similar across all three resolutions (Fig.3a). However, the WRF9K consistently showed lower precipitation values than expected (Fig.3b). This suggests that the underestimation of precipitation, rather than temperature differences, is the main cause of SWE bias in the lowest resolution model.

**Response:** It has been changed as suggested**.**

4 - From Discussion:

Original: "The large bias in the coarsest resolution also may be due to the incapability of the WRF9K to simulate the processes that are responsible for snow deposition and redistribution in mountainous areas that are characterized by heterogeneous snow distribution."

Can be improved: "The WRF9K model shows large errors primarily because of its resolution limitations. At 9km resolution, the model cannot accurately capture two key processes in mountainous areas: snow deposition and snow redistribution. These processes are especially important in mountains because snow distribution varies greatly over short distances."

**Response:** It has been changed as suggested**.**